

# A novel, cost-effective analytical method for measuring high-resolution vertical profiles of stratospheric trace gases using a GC-ECD

Jianghanyang Li[1,2], Bianca C. Baier[2], Fred Moore[1,2], Tim Newberger[1,2], Sonja Wolter[1,2], Jack Higgs[2],
Geoff Dutton[1,2], Eric Hintsa[1,2], Bradley Hall[2], and Colm Sweeney[2]

[1]Cooperative Institute for Research in Environmental Sciences, University of Colorado Boulder, Boulder, CO, USA, 80309
[2]Global Monitoring Laboratory, National Oceanic and Atmospheric Administration, Boulder, CO, USA, 80305

*Correspondence to*: Jianghanyang Li (jianghanyang.li@noaa.gov)

**Abstract.** The radiative balance of the upper atmosphere is dependent on the magnitude and distribution of greenhouse gases and aerosols in that region of the atmosphere. Climate models predict that with increasing surface temperature, the primary mechanism for transporting tropospheric air into the stratosphere (known as the Brewer-Dobson Circulation) will strengthen, leading to changes in the distribution of atmospheric water vapor, other greenhouse gases, and aerosols in this region. Stratospheric relationships between greenhouse gases and other long-lived trace gases with various photochemical properties (such as $N_2O$, $SF_6$, and chlorofluorocarbons) provide a strong constraint for tracking changes in the stratospheric circulation. Therefore, a cost-effective approach is needed to monitor these trace gases in the stratosphere. In the past decade, the balloon-borne AirCore sampler developed at NOAA/GML has been routinely used to monitor the mole fractions of $CO_2$, $CH_4$, and CO from ground to approximately 25 km above mean sea level. Our recent development work adapted a gas chromatograph coupled with an electron capture detector (GC-ECD) to measure a suite of trace gases ($N_2O$, $SF_6$, CFC-11, CFC-12, H-1211, and CFC-113) in the stratospheric portion of AirCores. This instrument, called the StratoCore-GC-ECD, allows us to retrieve vertical profiles of these molecules at high resolution (5-7 hPa per measurement). We then launched four AirCore flights and analyzed the stratospheric air samples for these trace gases. The results showed consistent and expected tracer-tracer relationships and good agreement with recent aircraft campaign measurements. Our work demonstrates that the StratoCore-GC-ECD system provides a low-cost and robust approach to measuring key stratospheric trace gases in AirCore samples and for evaluating changes in the stratospheric circulation.

## 1. Introduction

Monitoring the dry air mole faction of a suite of trace gases in the stratosphere will significantly improve our understanding of the stratospheric mean meridional circulation's (Brewer-Dobson Circulation, or BDC) response to the changing climate. The BDC is characterized by upwelling in the tropics, with upper and lower poleward branches and descent in the extra-tropics (Butchart, 2014; Holton et al., 1995; Garcia and Randel, 2008). Coupled chemistry climate models (CCMs) predict an acceleration of the BDC in response to increasing greenhouse gas abundances and surface



temperatures (Butchart et al., 2006; McLandress and Shepherd, 2009; Butchart, 2014; Garcia and Randel, 2008), with far-reaching implications for surface weather and climate (Randel et al., 2006; Forster and Shine, 2002; Gerber et al., 2012), recovery of the stratospheric ozone layer (Butchart and Scaife, 2001; Butchart et al., 2010) and potential impacts to surface air quality due to changes in stratosphere-to-troposphere ozone flux and the tropospheric oxidation capacity (Hegglin and

Shepherd, 2009). However, directly measuring the strength and variation of the BDC is difficult.

The mean age of air (AoA) in the stratosphere, derived from measurements of long-lived tracers therein, such as carbon dioxide ($CO_2$) and sulfur hexafluoride ($SF_6$, Andrews et al., 2001; Ray et al., 1999; Waugh and Hall, 2002), has been suggested to be an indicator of the BDC strength (Engel et al., 2009; Stiller et al., 2017, 2012). The measurement derived AoA can be compared with modelled AoA to investigate the model's performance in simulating the BDC. However, later

studies showed that the BDC is not the only factor controlling the mean AoA, as it is also affected by the mixing of air from the extra-tropics back to the tropics, i.e., recirculation (Dietmüller et al., 2017; Ploeger et al., 2015; Ray et al., 2014; Garny et al., 2014). Additional work suggested that the stratospheric dry mole fractions of some long-lived trace gases, such as nitrous oxide ($N_2O$), and several chlorofluorocarbon molecules (CFCs), including dichlorodifluoromethane (CFC-12), trichlorofluoromethane (CFC-11), 1,1,2-trichloro-1,2,2-trifluoroethane (CFC-113), and bromochlorodifluoromethane (halon

1211, or H-1211), could provide further constraints to help better understand stratospheric circulation and transport pathways of air into the stratosphere (Schoeberl et al., 2000; Strahan et al., 1999; Volk et al., 1996; Moore et al., 2014). This is because 1) photolytic destruction is the sole sink for these gases, 2) their photolytic destruction rates increase exponentially with altitude, and 3) the altitude-photolytic lifetime profiles for these trace gases are different (Moore et al., 2014). Therefore, observations of age tracers (e.g., $CO_2$ and $SF_6$) and other photolytic trace gases are needed to carefully monitor, examine,

and verify simulated stratospheric transport.

The lightweight balloon-borne observation system called the AirCore provides a low-cost approach to observing the composition of the stratosphere (Tans, 2009; Karion et al., 2010). High-quality *in situ* measurements of stratospheric air are rare, with those acquired from occasional high-altitude large-balloon ($>10^6$ $m^3$) and aircraft-based field campaigns since the 1980s still relevant today for diagnosing stratospheric composition and dynamical change (Andrews et al., 2001; Hall et al.,

1999; Pan et al., 2010; Laube et al., 2020). The cost of such field campaigns prohibits routine measurements of stratospheric air. The AirCore was developed at the NOAA Global Monitoring Laboratory (NOAA/GML) and has been widely used to measure $CO_2$, $CH_4$, and CO profiles in samples collected from the surface to the stratosphere. The AirCore consists of a long (approximately 100 meter), thin, coated stainless steel tube with one open end. The gas in the AirCore evacuates as it ascends with a balloon. After the balloon is cut away at 30-32 km above mean sea level (AMSL), the AirCore descends, and

collects ambient air. Due to the relatively low volumetric flow rate and small cross-section area of the AirCore, the mixing of air captured in the AirCore is limited to Taylor diffusion (Aris, 1956) and molecular diffusion, largely preserving the vertical structure of the atmosphere in the AirCore. After landing, the AirCore is automatically closed, preserving the collected air sample until laboratory analysis. As AirCores are usually analyzed shortly (less than 4 hours) after landing, the composition of air from the ground to the mid-stratosphere can be analyzed with only a small amount (less than 0.7 meter in both



directions, Karion et al., 2010) of diffusion and dispersion mixing acting on the sample, allowing us to retrieve vertical gradients of trace gases in air from the ground to the mid-stratosphere with significant fidelity at a relatively low cost (~$5000/profile).

The most common analytical approach for analyzing AirCore samples employs continuous flow gas analyzers to derive the vertical profiles of $CO_2$, CO, $CH_4$, and $N_2O$ (Karion et al., 2010; Laube et al., 2020; Membrive et al., 2017; Engel

et al., 2017). In this approach, the AirCore sample is pushed through one, or a series of, continuous-flow gas analyzer(s), during which the analyzer(s) measure the dry mole fractions of several gases (CO, $CO_2$, $N_2O$, and $CH_4$) at a relatively slow flow rate (approximately 30 mL min$^{-1}$) and measurement cadence (~0.45 Hz). These measurements are then combined with flight data (such as altitude, pressure, and temperature) to derive vertical profiles of measured trace gases with altitude using estimates of flow impedance due to flow resistance in a laminar regime as sample air moves along the length of tubing

(Tans, 2022). Although this method provides fast, high-resolution measurements of several essential trace gases, the continuous analyzers cannot directly measure other trace gases of interest for evaluating changes in the BDC (such as CFCs or $SF_6$), limiting the species measured from an AirCore sample. Mrozek et al. (2016) have designed a sub-sampling system that separates the AirCore samples into 25 mL aliquots, allowing for more detailed chemical and isotopic measurements using non-continuous flow analytical instruments. This method was then applied to measure the dry mole fractions of CFC-

11 and other trace gases in each sub-sample (Laube et al., 2020) and to investigate the mass-independent fractionation signals in $CO_2$ molecules (Mrozek et al., 2016). However, to date, only around ten stratospheric measurements have been sub-sampled from a 2 L volume AirCores. With the weight of NOAA unmanned free balloon payloads limited to 5.4 kg by the Federal Aviation Administration (FAA) in the United States, the method used by Mrozek et al. (2016) would provide lower vertical resolution and thus limited utility in resolving critical stratospheric gradients of these gases. Additionally,

NOAA's AirCore sampling program routinely deploys two samplers simultaneously, which currently restricts the total volume of each AirCore to less than 1 L. Therefore, an alternative approach is needed to measure the mole fractions of several critical trace gases in AirCores at a higher vertical resolution.

Here, we present a novel analytical method using a modified gas chromatograph coupled with an electron capture detector (GC-ECD) system to analyze the dry mole fractions of CFC-11, CFC-12, CFC-113, H-1211, $N_2O$, and $SF_6$ from the

stratospheric portion of AirCore samples (i.e., the top 25% of the sampler tubing) at high vertical resolution. We name this system the StratoCore-GC-ECD. The StratoCore-GC-ECD is designed to accomplish high-precision measurements of these six species using only ~4-5 mL of air sample per measurement from the stratospheric portion of AirCores while carefully registering the altitude information of each data point, allowing us to acquire high-resolution measurements of the vertical gradient of these trace gases from the tropopause to the mid-stratosphere. This analytical method provides high-resolution

vertical profiles of a suite of both AoA tracers and photolytic tracers in the stratosphere, offering the potential for long-term monitoring of these gases using a balloon-borne sampling package that is regulated under the same flight rules as those that apply to weather balloons. This methodology, coupled with the AirCore will provide us the flexibility to measure important stratospheric tracers at enhanced spatial and temporal resolution over current analytical methods. Such observations will



provide us with valuable information to monitor a suite of trace gases in the stratosphere long-term at low-cost, define

baseline stratospheric conditions for any perturbations in stratospheric composition due to future climate intervention

techniques, and provide observational evidence to detect and monitor changes in the BDC.

## 2. StratoCore-GC-ECD setup

### 2.1 Gas Chromatography in the GC-ECD system

The sample analysis portion of the StratoCore-GC-ECD system is adopted from the previous GC systems designed

and built for rapid, high-frequency *in situ* analysis on aircraft and large balloon platforms (Romashkin et al., 2001; Hintsa et

al., 2021; Elkins et al., 1996; Moore et al., 2003). Figure 1 displays a diagram of the StratoCore-GC-ECD system. The

analysis component of the system consists of a two-channel GC-ECD, which mimics the design of the UAS Chromatograph

for Atmospheric Trace Species (UCATS, Hintsa et al., 2021) and the *in situ* GC system used during the Lightweight

Airborne Chromatograph Experiment campaign (LACE, Moore et al., 2003). The GC system uses ultra-high purity nitrogen

gas ($N_2$) as a carrier gas. In each GC channel, a Valco 12-port 2-position valve (VICI, TX, USA) is used to switch between

sample loading (into two 1 mL sample loops) and injecting modes. Each analysis takes 120 seconds in this setup. Channel 1

uses a 10% dimethylsilicone (OV-101) pack column in the pre-column to separate CFC-12, H-1211, CFC-11, and CFC-113

in the sample, which subsequently passes through the main column and is analyzed by the ECD detector. A temperature

controller (model CNI16-AL, Omega, CT, USA) is used to control column temperature at 38 Celsius. The flow rate of

carrier gas in this channel is 70 mL per minute, and the pre-column is backflushed for 85 seconds at 100 mL per minute in

each analysis to remove the residual sample. Similarly, Channel 2 uses porous polymers (HayeSep® D) pack columns

followed by Molecular Sieve 5A to separate $SF_6$ and $N_2O$ (controlled at 110 °C), which are then analyzed by the second

ECD detector. The flow rate of carrier gas in Channel 2 is 70 mL per minute. Backflushing in channel two occurs after 55

seconds of each 120-second analysis at 100 mL per minute to remove any residual sample. In addition, a small flow of pure

$CO_2$ (0.2 mL per minute) is mixed into the ECD detector in Channel 2 as a dopant to minimize the matrix effect and improve

the ECD response to $N_2O$ (Fehsenfeld et al., 1981).

The StratoCore-GC-ECD system displays adequate analytical precisions suitable for measuring the dry mole

fractions of CFC-11, CFC-12, CFC-113, H-1211, $N_2O$, and $SF_6$ (hereafter referred to as target molecules) in the stratosphere.

A typical chromatography of the GC-ECD system is shown in Figure 2. The analytical repeatability of the GC-ECD for the

target molecules is evaluated by measuring gas cylinders with well-determined dry mole fractions of target molecules

multiple times, and the uncertainties are shown in Table 1. Considering the dry mole fractions of these species in the

stratosphere display wide ranges (50%-100% overall variations), such analytical precisions (<0.7%) of the GC-ECD should

be sufficient to understand the stratospheric variability of these species. A set of five gas mixtures in Aculife-treated

aluminium cylinders, spanning the range of expected stratospheric dry air mole fractions (20% to 100% of tropospheric



values) were prepared and used to calibrate the GC-ECD. Examples of the most recent calibration curves are shown in
       Figure 3.

## 2.2 Sample handling in the StratoCore-GC-ECD system

A sample handling apparatus is designed to analyze AirCore samples (Figure 1). Airborne *in situ* GC-ECD systems
typically use a high sampling flow (~100 mL per minute) to flush the sample loading system prior to analysis (Romashkin et
al., 2001; Hintsa et al., 2021; Elkins et al., 1996; Moore et al., 2003). However, the limited amount of AirCore sample (here,
       <250 mL of air) requires an alternative approach to load air into the GC-ECD. To achieve this goal, an in-house sample
       handling system was specially designed and built at NOAA/GML for capturing and injecting sample gas from the AirCore
       into the GC-ECD system. The flow path in the sample handling system is controlled by a Valco 6-port 2-position valve
       (VICI, TX, USA) and pushed by an in-house standard gas cylinder with dry mole fractions of target molecules that are well-
determined. The 6-port valve setup allows this "push gas" to also act as a calibration standard that can be directly injected
       into the GC-ECD periodically (through the bypass position, Figure 1). With each sample loading, the sample flow rate is
       carefully controlled by a mass flow controller (Mykrolis, MA, USA) to maintain a stable pressure profile and constant flow.
       The flow rates during the sample loading process are then accurately measured by a mass flow meter (Omega, CT, USA) at
       1 Hz at the outlet of the StratoCore-GC-ECD system. The flow measurements associated with each sample loading process
are integrated to calculate the total volume of air coming out of the AirCore for each measurement. The total sample volume
       data are used for registering the location of each measurement along the length of the AirCore (given a known total volume
       of the AirCore) which is a crucial step for registering the GC-ECD measurements with altitude (Tans, 2022).

## 2.3 In-lab testing of the StratoCore-GC-ECD system

Two tests were conducted to the StratoCore-GC-ECD system to 1) examine potential contaminations of target
molecules from the AirCore, 2) evaluate the mixing of air samples along the direction of flow during analysis, and 3) assess
       the accuracy of AirCore volume registration. The AirCore used in the tests of this study shares the same material and coating
       technology as the AirCores currently used by NOAA/GML for measuring atmospheric vertical profiles of $CO_2$, $CH_4$, and CO
       (Baier et al., 2018; Karion et al., 2010). The test AirCore consists of a 30 m long 304-grade stainless steel tubing, with an
       outer diameter of 0.32 cm and an inner diameter of 0.29 cm. The inside wall of the tubing was treated with SilcoNert coating
(SilcoTek Corp., PA, USA). To examine the potential contaminations from the wall of the AirCore, two experiments were
       conducted. In the first experiment, the AirCore was flushed with zero-grade air, stored overnight (~14 hours), then analyzed
       using the StratoCore-GC-ECD system, during which a standard gas of typical tropospheric composition was used as a "push
       gas". The second experiment was the opposite of the first: the AirCore was first flushed with the standard gas, stored
       overnight (~14 hours), then zero air was used as a push gas to analyze the AirCore using the StratoCore-GC-ECD system.



The results of both experiments, shown in Figure 4, suggest that 1) there is no observable contamination from the AirCore tubing, 2) the mixing of air samples during analysis is insignificant and can be modelled, and 3) the accuracy of volume registration (and therefore, altitude registration when measuring an AirCore sample) by the StratoCore-GC-ECD system is accurate to within 0.6%. In the first experiment, after the AirCore stored zero air for 14 hours, the dry mole fractions of all target molecules were below the StratoCore-GC-ECD detection limit for these species in the entire AirCore;

similarly, in the second experiment, none of the target molecules measured demonstrated any significant change in value after 14 hours of storage. Considering the storage time of actual AirCore samples (the time from AirCore landing to analysis) is usually within 4 hours, the results suggest that the AirCore sampler surfaces do not act as a surface contaminant for samples during regular AirCore flights.

The two experiments also demonstrated limited mixing of the sample during the analysis. The push gas used in both

experiments differed from the gas in the AirCore, defining the transition between the AirCore sample and the push gas after each analysis. The measurements display sharp transitions from the sample to the push gas within 1-2 injections (<10 mL of air). These abrupt transitions indicate that due to the carefully controlled flow during the sample loading process and minimized pressure drop during valve switching, the sample mixing during the analysis in the AirCore is minimal. Here, we used a simple mixing model to estimate the molecular and Taylor diffusion between the push gas and the sample gas in the

AirCore:

$$X_{rms} = (2 * D_{eff} * t)^{0.5},$$    Equation 1

in which $X_{rms}$ is the root-mean-squared diffusion distance, $t$ is time, and $D_{eff}$ is an effective diffusivity incorporating both the molecular diffusion and the Taylor diffusion:

$$D_{eff} = D + \frac{a^2 v^2}{48D}.$$    Equation 2

In Equation 2, $D$ is the molecular diffusion coefficient, $a$ is the tube radius, and $v$ is the average air velocity. Using the equations, we modelled the diffusion between the push gas and the sample gas during the analysis period (~1 hour), shown in Figure 4C. The modelled $X_{rms}$ on the back end of the AirCore was 37.4 cm (equivalent to 1.4 mL of air), in line with the observed sharp transition between the push gas and the sample gas. Therefore, we suggest that the flow in the StratoCore-GC-ECD system during sample analysis remains a rigid "slug flow" moving along the system.

Additionally, the results from the two experiments also demonstrated high accuracy of volume registration by the StratoCore-GC-ECD system. Using the mass flow meter in the StratoCore-GC-ECD system, we registered each data point to the volume of the AirCore. The measured volume of the AirCore by the StratoCore-GC-ECD system is defined as the midpoint of the transition between the AirCore gas and the push gas. In the meantime, we carefully measured the true volume of the AirCore multiple times by weighing the amount of water needed to fill the entire AirCore. The total volume of

the AirCore measured by the StratoCore-GC-ECD system agrees with the actual volume of the AirCore within ± 1 mL, suggesting the volume measurements by the StratoCore-GC-ECD system are accurate to within ±0.6%.



Another set of tests was also conducted to further verify that using a simple 1-D diffusion model, we can quantify the diffusion of the sample in the AirCore during storage. In these tests, the AirCore was filled with two alternating slugs using two calibrated dry standard gas cylinders with different dry mole fractions of all the target molecules and $CO_2$, $CO$,

and $CH_4$. The transitions between slugs in the filling process (Karion et al., 2010) were directly measured by a continuous gas analyzer (G2401-m, Picarro, CA, USA) and used as a baseline condition. Subsequently, the AirCore was closed, quickly connected to the StratoCore-GC-ECD system then analyzed immediately. The results are shown in Figure 5A: the transitions between slugs observed by the StratoCore-GC-ECD system display good agreement with the baseline condition, again suggesting mixing induced by StratoCore-GC-ECD's sample loading process is negligible. In the next test, the AirCore was

filled with two alternating slugs (the same as the previous experiment), then stored for 26 hours before being analyzed by the StratoCore-GC-ECD. After the 26-hr storage, the mixing between slugs was significant (Figure 5B). Assuming a 1-D diffusion model along the length of the AirCore, we used Equation 1 to model the molecular diffusion inside the AirCore during the storage. The modelled diffusive mixing between slugs agrees with the observed mixing (Figure 5B), suggesting that the horizontal mixing between AirCore landing and sample analysis can be calculated and accounted for within the

uncertainty in the AirCore altitude registration.

### 3. Balloon-borne AirCore flights

Four balloon-borne AirCore test flights were conducted to retrieve the vertical profiles of CFC-11, CFC-12, CFC-113, H-1211, $N_2O$, and $SF_6$, in the stratosphere. The four test flights were launched in Eastern Colorado on September 8, 2021 (Flight 1), November 16, 2021 (Flight 2), March 31, 2022 (Flight 3), and August 9, 2022 (Flight 4), respectively. A

3000 g weather balloon filled with helium carried the flight string on each flight to ~ 29.5 km AMSL. The payload package in the test flights is identical to the routine AirCore flights of the ongoing AirCore Program at NOAA/GML (Figure 6). Each package contains a parachute, a cutter that can be remotely controlled to release the balloon and open the parachute, a GPS for real-time tracking of the flight, an Automatic Dependent Surveillance-Broadcast (ADS-B) transponder, a radiosonde (InterMet systems iMet-1 RSB, MI, USA) for recording latitude, longitude, altitude, temperature, atmospheric pressure,

relative humidity, and wind speed, and two AirCores. The two AirCores used in each flight are identical to ensure the sampling processes of both AirCores are the same and confirm that there is no contamination to either of the AirCore sampler. In Flights 1-3, the AirCores consisted of 109-meter-long tubing, with an outer diameter of 0.32 cm and inner diameter of 0.29 cm (total volume: 600 mL), with two valves placed on both ends (top and bottom valves). The AirCores used in Flight 4 (August 9, 2022) were optimized for stratospheric air sampling (see discussions below). The AirCores used

in Flight 4 consisted of two segments: the bottom portion (open end when sampling) was a 21-meter tube with an inner diameter of 0.58 cm, and the top part was a 28-meter tube with an inner diameter of 0.29 cm (740 mL total volume with approximately 25% in the thinner tubing). Before the flight, each AirCore is insulated using a polymer foam package, wrapped with plastic wrap to minimize damage in the field, and covered by a custom-made bag using high-strength,



lightweight Dyneema composite fabric. A data logger (Arduino, MA, USA) placed next to the AirCore inside the polymer
foam recorded the coil's temperature at multiple locations, latitude, longitude, altitude, and atmospheric pressure at 1 Hz.

Hours before each flight, AirCores are flushed with special gases to distinguish between the residual fill gas not
evacuated from the AirCore during flight and atmospheric sample during analysis. The AirCores used for $CO_2$, $CH_4$, and CO
measurements (by continuous analyzers) are flushed with an air mixture with a high CO dry mole fraction (1765 ppb) and
known $CO_2$ and $CH_4$ dry mole fractions. The AirCores analyzed on the StratoCore-GC-ECD system are flushed with zero-air
containing elevated H-1211 (6.6 ppt). We selected H-1211 as the tracer for the remaining fill gas because of the rapid
photochemical destruction of H-1211 in the lower stratosphere: atmospheric models and *in situ* aircraft observations
(Portmann et al., 1999; Papanastasiou et al., 2013; Moore et al., 2014; Elkins et al., 2020) showed H-1211 is destroyed
entirely above 25 hPa (~20-22 km AMSL). Since the topmost sampling altitude of our AirCore is higher than 25 km AMSL,
H-1211 is an ideal tracer for separating the residual fill gas in the AirCore with the air sample collected above 24-25 km
AMSL. After the AirCores are thoroughly flushed, the bottom valves are kept closed until minutes before the flight to
minimize potential contamination from ambient air.

The flight trajectories of all four test flights are shown in Figure 7. The balloon setup is designed such that the
payload for all the flights should have similar ascent and descent processes: the balloon carries the payload to 28.9 km
AMSL (10 hPa) at ~ 6 m/s, then a balloon cutter is activated to release the payload from the balloon. After the balloon
cutaway, the parachute is deployed, and the payload descends for approximately 42 minutes, during which the AirCores
passively collect ambient air. One exception was Flight 3: after the balloon cutaway, the parachute did not fully open,
resulting in a faster descent (landing in 18 minutes). After landing, the bottom valve on the AirCore closes automatically
after ~30 seconds to minimize sample loss and potential contamination and loss of sample (due to warming). The AirCores
were quickly transferred back to the lab for analysis. In each flight, one of the AirCores was analyzed by a Picarro-2401
continuous gas analyzer for $CO_2$, $CH_4$, and CO dry mole fractions. The other AirCore was analyzed by the StratoCore-GC-
ECD (only the stratospheric portion was analyzed). After the analysis, the filling process of the AirCore during descent is
modelled using the meteorological data, and the AirCore fluid dynamic program (Tans, 2022). The modelled results are then
used to register the sample measurement time series with the altitude at which each sample was collected to derive the
vertical profiles of all the trace gas measured by both instruments.

The AirCore dimensions have a significant impact on the sampling efficiency in the stratosphere (e.g., Membrive et
al., 2017). The AirCores used in Flight 4 had wider tubing (0.58 cm inside diameter) at the bottom (open end) and thinner
tubing on top (0.29 cm inside diameter), while the AirCores used in Flights 1-3 consisted of one piece of thin tubing (0.29
cm inside diameter). As a result, their sampling efficiencies in the stratosphere are drastically different. Using the fluid
dynamic model described in Tans (2022), we modelled the outflow and inflow of AirCores during each flight, shown in
Figure 8. The model suggests that after the descent, the volumetric inflow during flight 4 (Figure 8A) increased much more
rapidly compared to flights 1-3. The higher inflow in flight 4 is due to the larger diameter on the bottom portion of the
AirCores used in flight 4, which produces a much smaller pressure gradient along the length of the tubing, making it easier



for air to enter the sampler. Therefore, the stratospheric sampling efficiency, i.e., the ability of the AirCore to collect
stratospheric air (Figure 8B), of flight 4 was significantly higher than those of flights 1-3. The increased flow rates in flight 3
(compared to flights 1 and 2) were caused mainly by its fast descent, creating a large pressure gradient at the inlet of the
AirCores. However, the combination of high flow rates and short descent time (60% quicker than other flights) in flight 3
still resulted in higher flow resistance in the AirCore which reduced sampling efficiency. Indeed, the model shows that
compared to the AirCores in flights 1 and 2, the AirCores in flight 4 are much closer to pressure equilibrium between the
close and open ends of the AirCore (Figure 8C), while the AirCores in flight 3 displayed the most significant imbalance open
and closed end during descent. This is demonstrated by the observed pressure differential between the open and close ends of
AirCore. The pressure differential of the entire AirCore in every flight was measured at 1 Hz. Using the model output, we
also calculated the overall pressure differential and compared it with the measurements. The modelled pressure differential
(Figure 8D) time series agrees with measurements in all the flights (data not shown in the figure): during the AirCore
descent, the root-mean-square error (RMSE) between model output and measurements is less than 0.6 hPa (corresponding to
approximately 240 m at 28 km AMSL and 20 m at 12 km AMSL), suggesting the model had successfully reproduced the
sampling process of AirCores. The AirCores used in flight 4 showed a much smaller pressure differential compared to the
AirCores used in flights 1-3, again highlighting the higher stratospheric sampling efficiency of the modified AirCore
dimensions.

The StratoCore-GC-ECD analysis of AirCores from the four test flights yielded high vertical resolution profiles of
target molecules, agreeing well with their predicted stratospheric photochemical loss processes. For all the target molecules
measured by the StratoCore-GC-ECD system (Figure 9A-F), analyzing the 600 mL AirCores (in flights 1-3) produces 31 to
38 stratospheric measurements from each AirCore, equivalent to one measurement every 5-7 hPa. The larger dual-diameter
AirCore used in flight 4 yielded 50 measurements in the stratosphere with a resolution of 4.5 hPa per measurement
(corresponding to approximately 1.6 km per measurement at 28 km AMSL and 0.14 km per measurement at 12 km AMSL).
All the trace gases measured here display dynamic ranges in the stratosphere, including age tracer ($SF_6$) and photolytic
tracers ($N_2O$, CFC-12, CFC-113, CFC-11, and H-1211). The decrease in dry mole fractions of the photolytic tracers with
altitude can be explained by their stratospheric photochemical properties (Portmann et al., 1999; Moore et al., 2014):
compared to mean tropospheric values, the average loss (in %) of each photolytic tracer at the 650 K isentrope for the four
test flights are 58±5%, 61±4%, 72±3%, 93±3%, and 100% for $N_2O$, CFC-12, CFC-113, CFC-11, and H-1211, respectively.
These values agree well with the relative stratospheric loss of the tracers via photolysis: at any given altitude in the
stratosphere, the photolysis lifetime of each molecule, from longest to shortest, are $N_2O$ > CFC-12 > CFC-113 > CFC-11 >
H-1211. In addition, the high-resolution analysis from the StratoCore-GC-ECD systems captured temporal stratospheric
variability on scales of days to weeks, such as a positive excursion of all measured molecules at 75-85 hPa in Flight 2, or the
variable dry mole fractions of all molecules at 60-180 hPa in Flight 3 (Figure 9). Similar structures were also observed in the
$CO_2$ and $CH_4$ profiles obtained from the continuous analyzers using the other AirCore on the exact flight string. Therefore,
the observed variability in the AirCore profiles is unlikely to originate from artifacts during the sampling or measurement



processes but reflects some short-term atmospheric conditions that might be developed from episodic stratospheric dynamic events. The observed variability which is significantly larger than detection limits and calculated mixing in the AirCore suggests that the StratoCore-GC-ECD system allows us to obtain high vertical resolution observations of the stratosphere.

The tracer-tracer relationships in the profiles collected to date show agreement with *in situ* observations from previous flight campaigns within analytical uncertainties. The relationships between different trace gases are shown in Figure 10 and compared with *in situ* aircraft measurements using the UAS Chromatograph for Atmospheric Trace Species (UCATS, Hinsta et al., 2021) during the NASA Dynamics and Chemistry of the Summer Stratosphere (DCOTSS) campaign at Kansas, United States in Summer 2021. The AirCores collected samples from a higher altitude (25 – 28 km AMSL),

where there is more aged air and more pronounced photolytic loss of trace gases compared to the ER-2 research aircraft (up to 21 km AMSL). Thus, data from StratoCore-GC-ECD measurements offer a larger observational range relative to the aircraft measurements. The relationships between CFC-11, CFC-12, and H-1211 measured from AirCores agree with those of UCATS (Figure 10A, B). For CFC-113, the StratoCore-GC-ECD measurements generally agree with UCATS measurements with a small but consistent (1-2 ppt) discrepancy (Figure 10C). We speculate this discrepancy originates from

different analytical methods used to calibrate working standards for the two measurements: the standards used in the UCATS measurement were calibrated using a GC-ECD. In contrast, the standards used in StratoCore-GC-ECD were calibrated using a GC-MS. A previous study showed a 1.3 ppt offset in CFC-113 measurements between a GC-ECD and a GC-MS (Rhoderick et al., 2015). It is possible that the observed discrepancy between StratoCore-GC-ECD and UCATS measurements reflected a similar offset. Future studies are needed to understand the origin of this offset. $SF_6$ measurements

from StratoCore-GC-ECD have been corrected to account for their growth in the troposphere using the global average growth rate of tropospheric $SF_6$ in 2021 (latest data), and the corrected $N_2O$ - $SF_6$ relationship also shows general agreement with UCATS data (Figure 10D) with a small offset. We speculate that this small offset might originate from the uncertainty in estimating the growth of $SF_6$. One exception is the upper-most AirCore sample in flight 3 (with the lowest $SF_6$ and $N_2O$ dry mole fractions), which is slightly different from the other three flights. We suggest this might be due to the short-term

stratospheric transport variability on a time scale of several days to weeks. This variability is most likely driven by a combination of seasonal changes in wave activity, quasi-biennial oscillation (QBO), El Niño–Southern Oscillation (ENSO), and other short-term, episodic events. Mapping this variation between AoA and photolytic loss-dominated tracers allows us to investigate these drivers of stratospheric dynamics further but is outside the scope of this analysis. As we accumulate additional data in further flights, we can likely distinguish between these short-term variations and long-term changes driven

by climate change.

## 4. Conclusions

The StratoCore-GC-ECD system, with a specially designed AirCore sample handling system (capable of injection of 4-5 mL of air for each analysis), can measure a suite of long-lived trace gases ($N_2O$, $SF_6$, CFC-11, CFC-12, H-1211, and



CFC-113) from AirCore samplers with analytical precisions below 0.7% for all gases. AirCore samplers designed with
dimensions specially optimized for stratospheric sampling can obtain high-resolution vertical profiles of the trace gases from
the tropopause to 28 km AMSL. Four test AirCore flights were conducted in eastern Colorado from Fall 2021 to Summer
2022, with AirCores analyzed by the StratoCore-GC-ECD system. The results showed good agreement with model
predictions and aircraft in situ measurements, suggesting that the StratoCore-GC-ECD system provides a robust, and low-
cost approach for observing the chemical compositions of the stratosphere. In the future, this system will be applied for
regularly monitoring the change of these trace gases in the stratosphere, providing additional observational constraints on
global climate models in a changing climate. We suggest that the sample handling system of the StratoCore-GC-ECD can be
adapted to other analytical techniques to allow even more measurements (such as isotopic measurements) from AirCore
samples in the future.

**Data availability**

Data used in this work is available at: https://doi.org/10.15138/VA4C-CY20 (Li et al., 2023).

**Author contribution**

Conceptualization: J. L., F. M., and B. C. B.; Data curation: J. L., B. C. B., F. M., T. N., S. W., J. H., G. D., E. H., and B. H.;
Formal analysis: J. L., F. M., and B. C. B.; Funding acquisition: F. M., and B. C. B.; Investigation: J. L., B. C. B., F. M., T.
N., S. W., J. H., G. D., and E. H.; Methodology: J. L., B. C. B., F. M., T. N., S. W., J. H., G. D., and C. S.; Project
administration: B. C. B., C. S., and B. H.;  Resources: B. C. B., F. M., G. D., C. S., and B. H.; Supervision: B. C. B., C. S.,
and B. H.; Validation: B. C. B., F. M., and E. H.; Visualization: J. L., and F. M.; Writing – original draft preparation: J. L.;
Writing – review & editing: J. L., B. C. B., F. M., T. N., S. W., J. H., G. D., E. H., C. S., and B. H.

**Competing interests**

The authors declare that they have no conflict of interest.

**Acknowledgements**

This research is supported in part by NOAA cooperative agreements NA17OAR4320101 and NA22OAR4320151. We thank
funding support from NOAA's Earth's Radiation Budget Initiative, NOAA CPO Climate & CI (Grant #03-01-07-001),
NASA grants 80ARC019T0011 and 80HQTR21T0076.



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



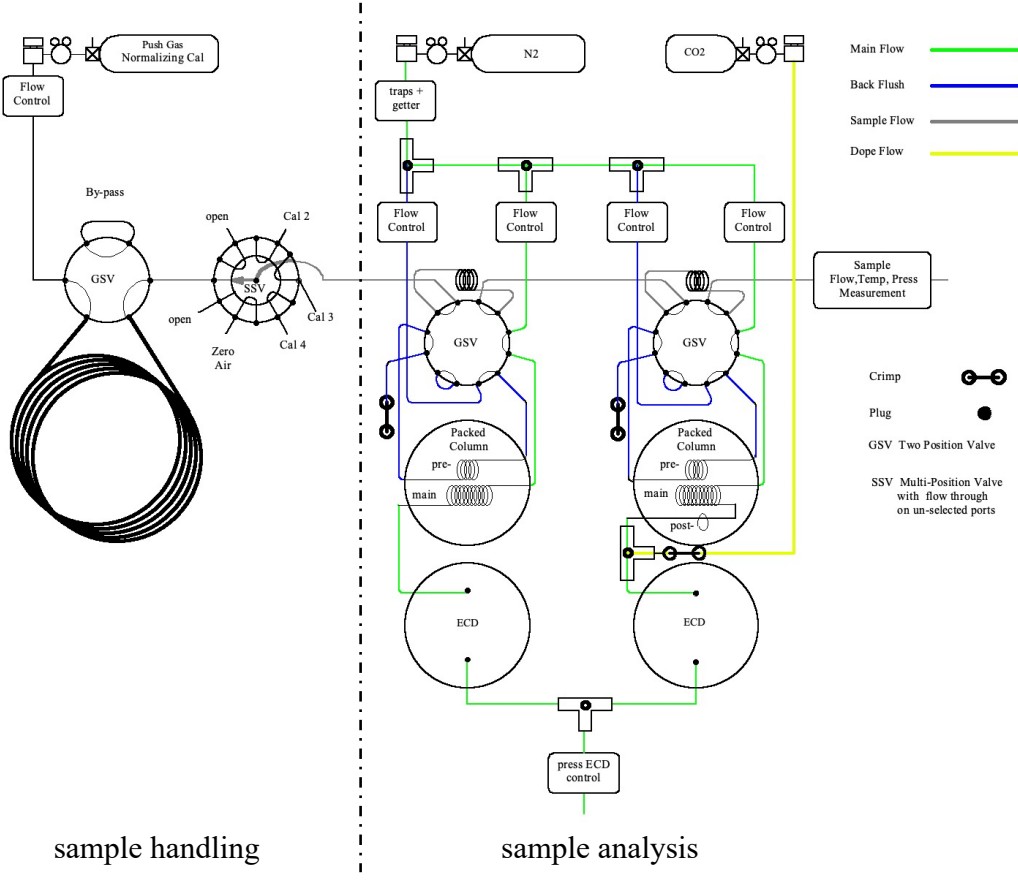

**Figure 1: Simplified sketch of the StratoCore-GC-ECD system. Dashed line marks the boundary between sample handling system and sample analysis system. The left side of the dashed line is the sample handling system that carefully injects the sample from AirCores (4-5 mL per analysis) into the GC-ECD. The right side of the dashed line is the sample analysis system that measures the mole fractions of six trace gases of each injection.**





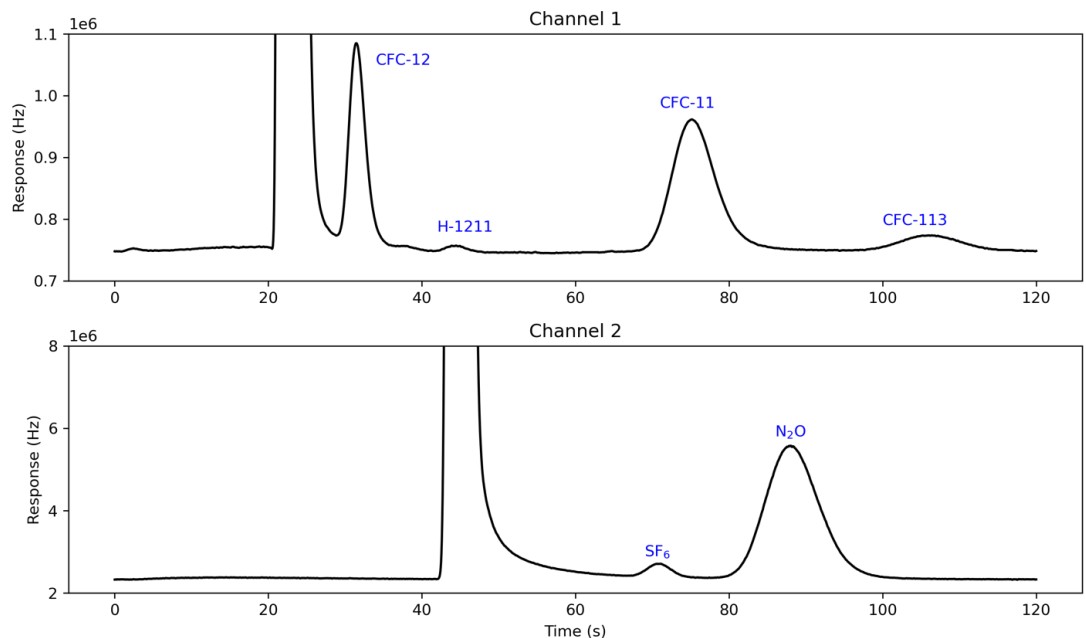


**Figure 2: A typical chromatograph from StratoCore-GC-ECD analysis. The x-axis is the retention time of each analysis, the y-axis is the response of the ECD. The top panel is the response of channel 1 (analyzing the mole fractions of CFC-12, H-1211, CFC-11, and CFC-113) and the bottom panel is the response of channel 2 (analyzing the mole fractions of N₂O and SF₆).**



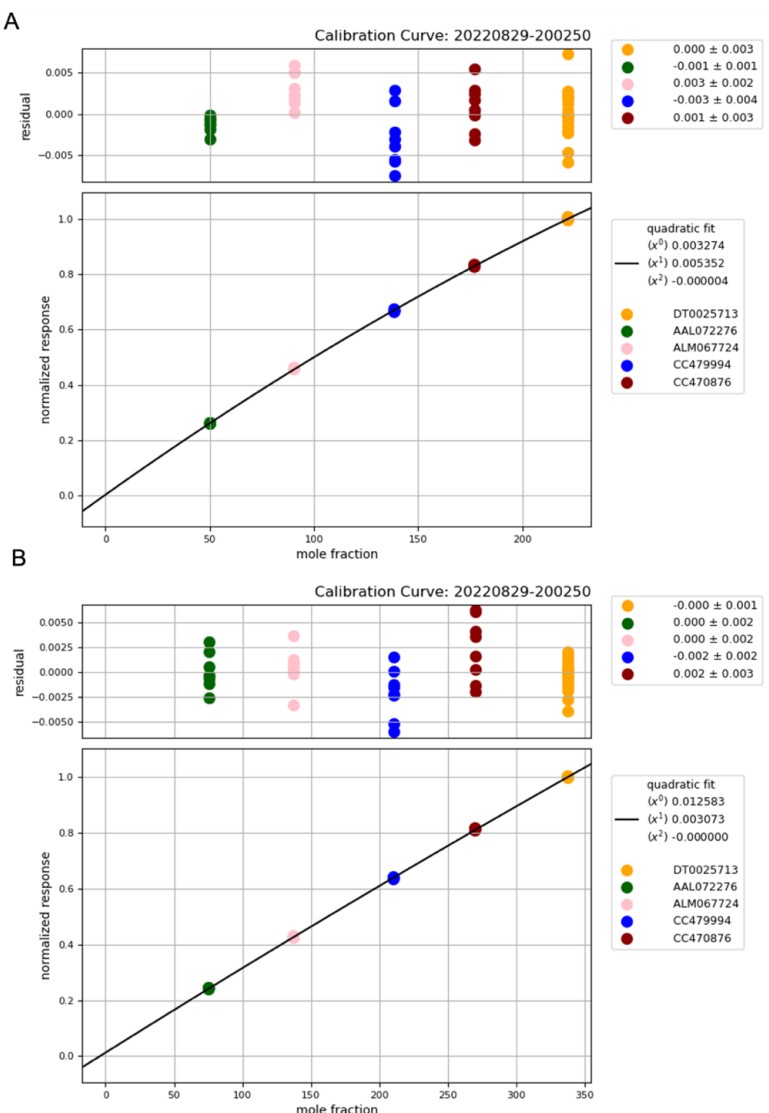


**Figure 3: Examples of calibration curves (A: CFC-11; B: N$_2$O) generated by analyzing five standard tanks using the StratoCore-GC-ECD system. Each color represents a different calibration tank, and each tank was measured a total of seven times. In each panel, the upper figure shows the relative residual mole fraction (unitless) between the measured value and the true curve, and the legend shows the mean residual of each tank; the lower figure shows the calibration curve and the parameters of the quadratic fit**
**function, and the legend shows the standards used.**





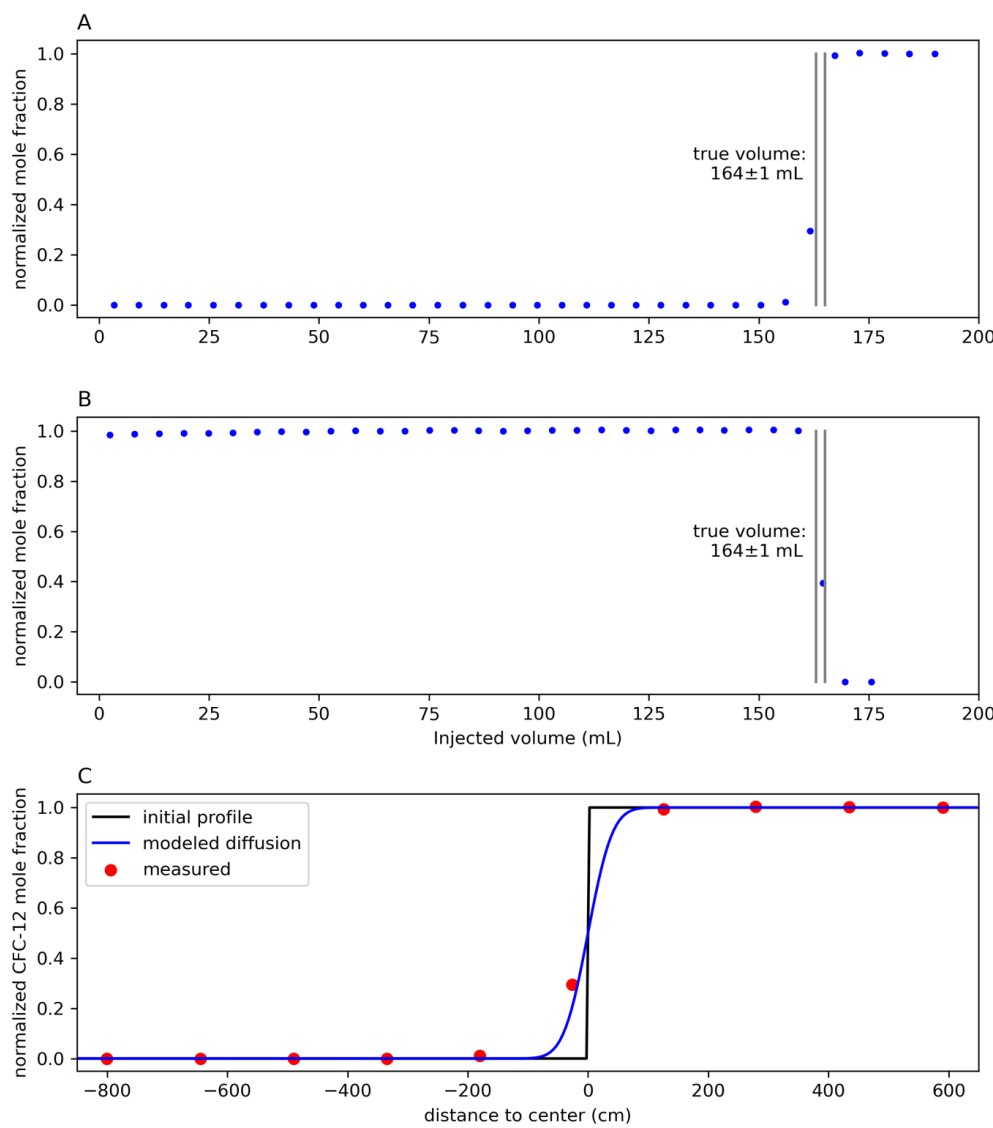

**Figure 4: Results of StratoCore-GC-ECD flow-through tests from AirCore sample for CFC-12 as an example. A: filling AirCore with zero air, then analyzing the AirCore using StratoCore-GC-ECD with air with species of tropospheric mole fractions as the push gas. B: filling AirCore with tropospheric air, then analyzing the AirCore using StratoCore-GC-ECD with zero air as push gas. C: modelled diffusion at the AirCore gas–push gas boundary during AirCore analysis after 12 mins/hours.**





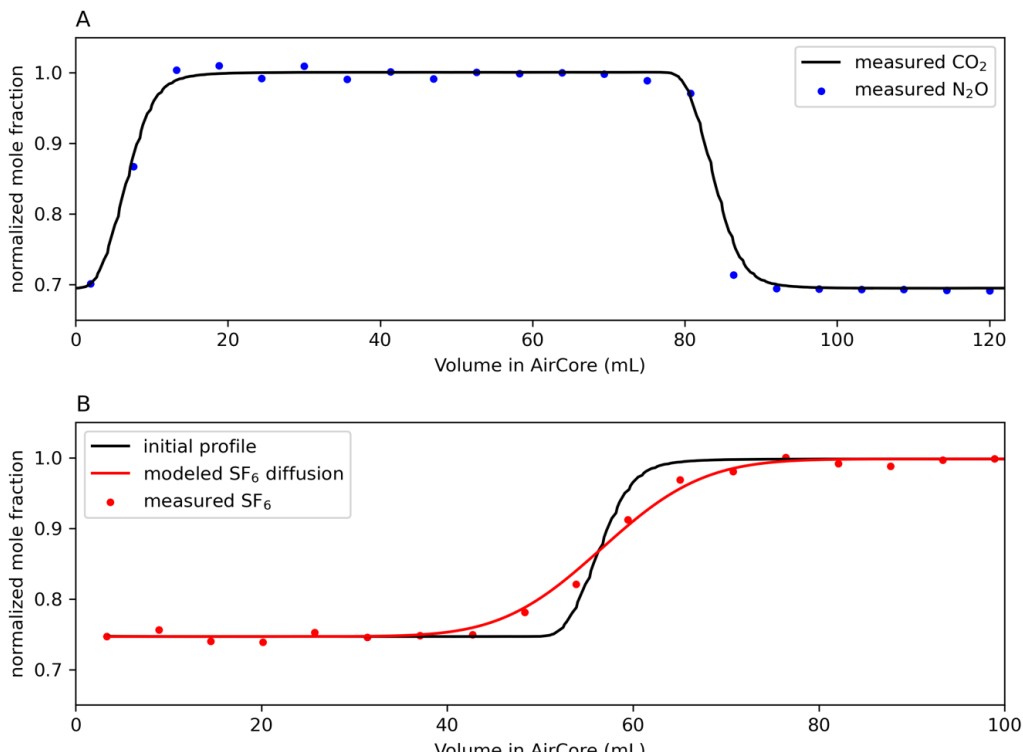

**Figure 5: A: results from the alternating-slug test. The AirCore was filled by two alternating slugs (with normalized dry mole**
**fractions of CFC-11 of 0.7 and 1, respectively), then immediately analyzed by StratoCore-GC-ECD. The black line represents the**
**transition between two slugs measured by the continuous flow analyzer; blue points are StratoCore-GC-ECD measurements. B:**
**Results of the storage test using the same alternate slugs. The black line represents the transition between two slugs when the**
**AirCore is being filled; the red line represents the modelled 1-D diffusion after 26-hour storage; the red dots are observed profile**
**measured by the StratoCore-GC-ECD.**






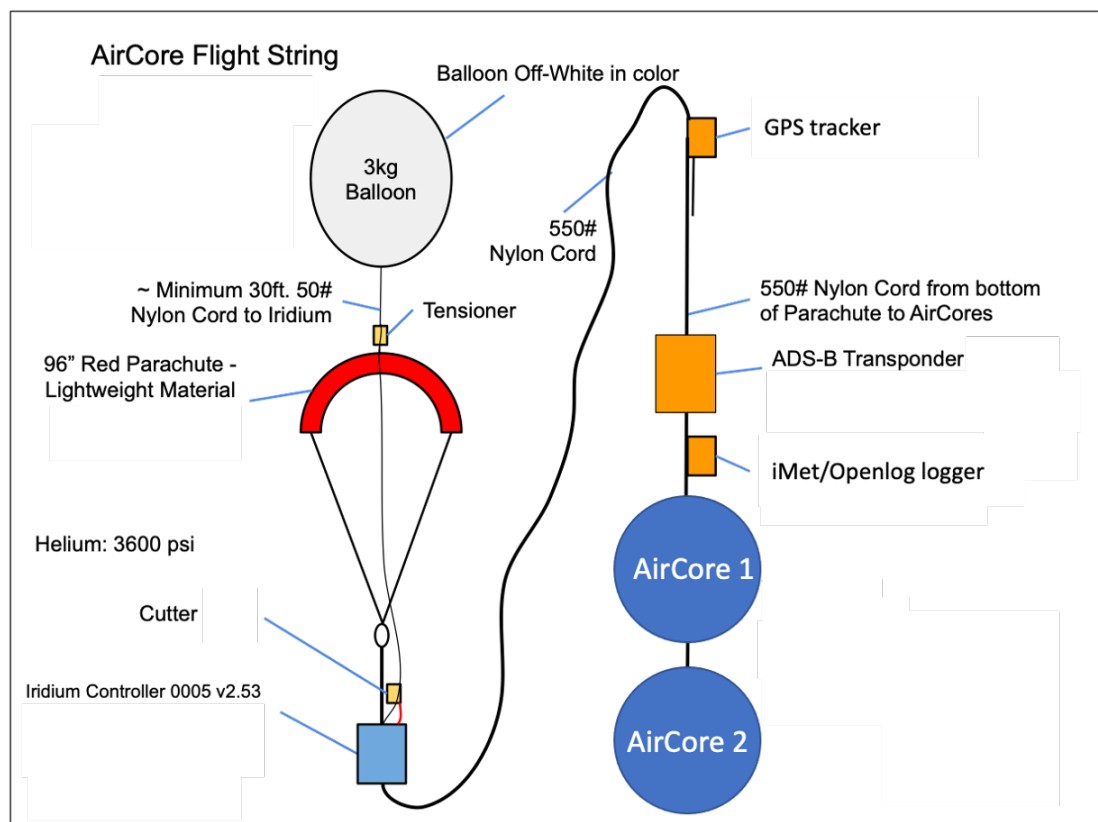

**Figure 6: Components of a typical NOAA AirCore flight train.**



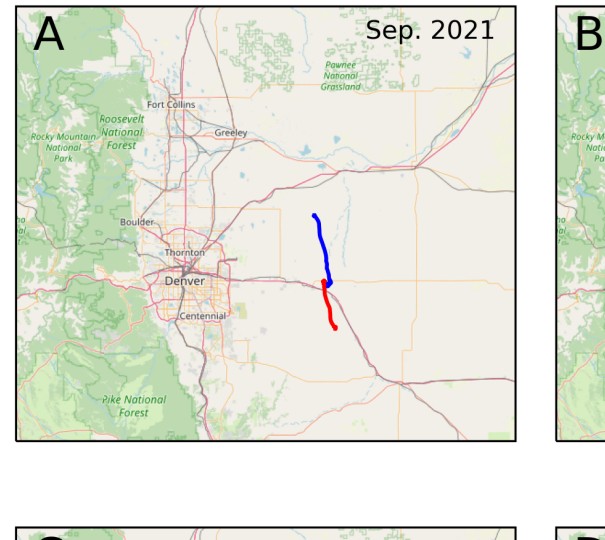

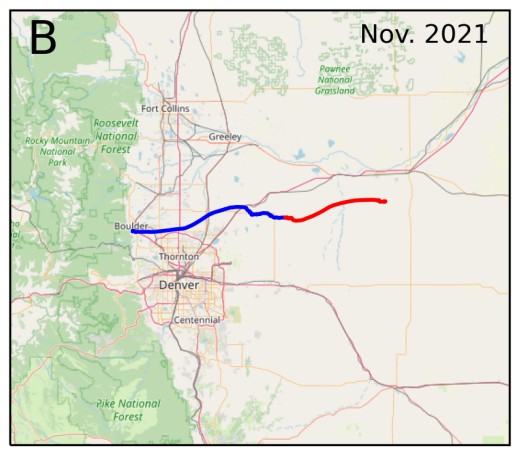

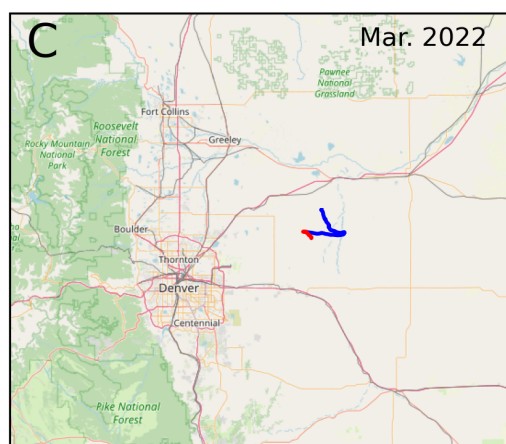

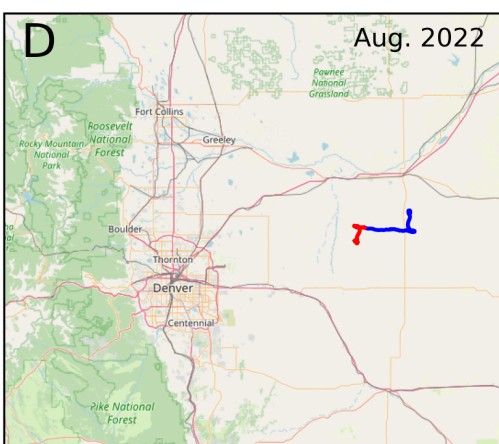

**Figure 7: Flight trajectories of the four test flights in Eastern Colorado, USA. Panels A-D represent flights 1-4. The blue lines represent the payload ascent. The red lines represent the payload descent. Base map is provided by OpenStreetMap.**

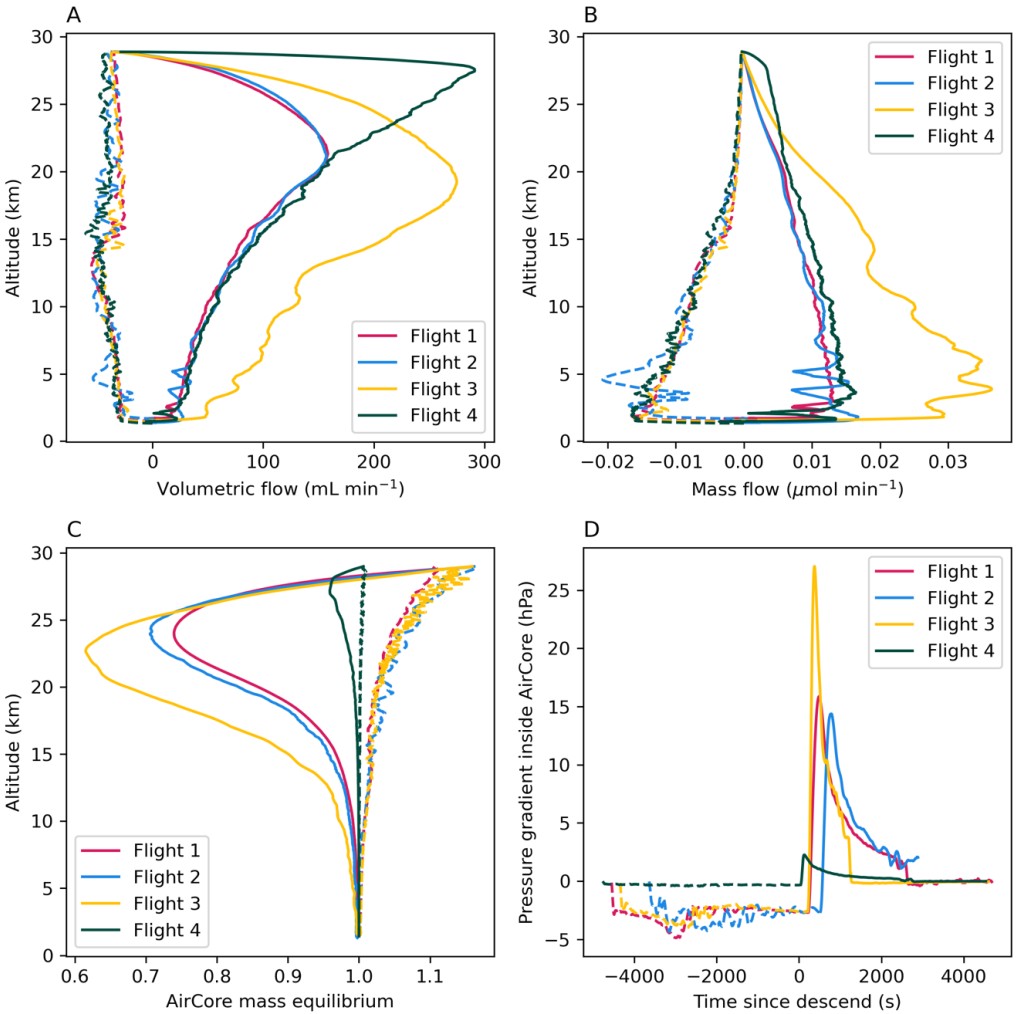

**Figure 8: Modelled fluid dynamics of the AirCore filling process. Each line represents one flight, the dashed portion of each line represents the ascent, and the solid part represents the descent. A: the modelled volumetric flow (in mL min⁻¹) into the AirCores during the entire flight, plotted against altitude; B: the modelled mass flow (in μmol min⁻¹) into the AirCores; C: the AirCore mass equilibrium ratio (actual air mass divided by equilibrium air mass in the AirCore) during the entire flight: a mass equilibrium ratio equal to 1 means the air inside the AirCore reaches equilibrium with ambient air, a ratio lower than 1 means the air inside the AirCore is depleted and vice versa; D: time series of the modelled pressure gradient across the entire AirCore.**

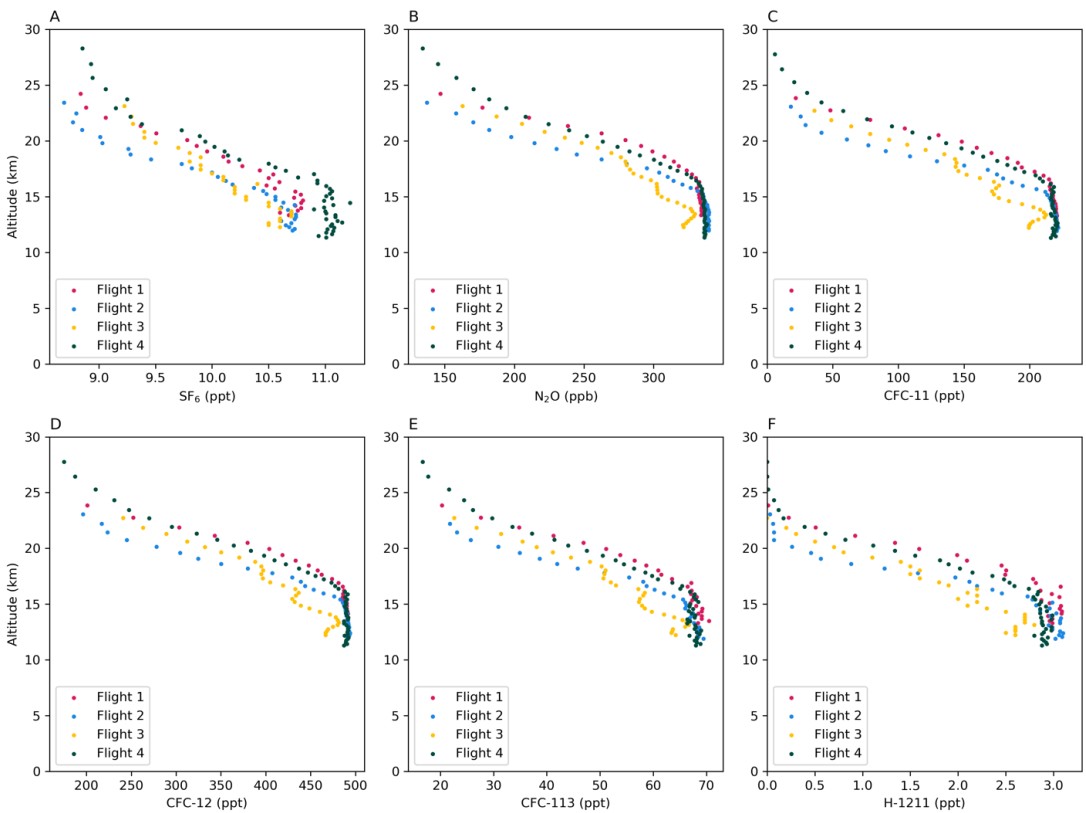

**Figure 9: Vertical profiles of AirCore trace gas dry mole fractions measured by the StratoCore-GC-ECD in the test flights from the tropopause to 25-28 km AMSL. Each color represents one of four total flights. A: $SF_6$, B: $N_2O$, C: CFC-11, D: CFC-12, E: CFC-113, F: H-1211.**






**Figure 10: Relationships between different molecules measured by the StratoCore-GC-ECD for Flights 1-4. Grey points are aircraft measurements acquired from the 2021 NASA DCOTSS campaign over the central United States and Canada. *AirCore SF$_6$ data in panel D were corrected (based on the global average growth rate of SF$_6$ in 2020-2021) to account for the growth of SF$_6$ in the atmosphere.**



**Table 1: Analytical repeatability of the StratoCore-GC-ECD.**

| Molecule | Analytical repeatability |
|---|---|
| CFC-11 | ±0.1 % |
| CFC-12 | ±0.1 % |
| CFC-113 | ±0.25 % |
| H-1211 | ±0.7 % |
| $SF_6$ | ±0.25 % |
| $N_2O$ | ±0.25 % |