# Peer review of "A novel, cost-effective analytical method for measuring highresolution vertical profiles of stratospheric trace gases using a GC-ECD"

_EGUsphere, 2023_

## Author Comment (AC1)

Overall Summary:

This paper is within the scope of AMT and should be published. There are some confusing sections, discussed further below, but overall the paper does a nice job of explaining a new technique that will improve our understanding of stratospheric circulation and add important measurements for long-term climate relevant gases. They prove that this technique works well and appears easy to add on to already existing infrastructure, and is cost-effective. There is nice agreement with aircraft measurements and any discrepancies are reasonably explained. I look forward to seeing more StratoCore-GC-ECD data in the future. Given that only one section of this paper needs restructuring, it should be accepted with minor revisions.

General comments:

The main section of the paper that needs improvement is section 2.3 and relevant figures. Section 2.3 was the most confusing of the entire paper and also critical to the verification of this new system. It was hard to follow the steps of the different experiments. In part this was a challenge because of the different language used (test, experiment, etc) along with the two tests (but maybe more?) to study three scientific objectives. This section is important and a restructure would be helpful for the reader to follow the logic. For example, line 160 mentions 'both experiments' and then line 192 mentions 'another set of tests'. Was that related to the first two tests mentioned at the start of the section? If so, set doesn't make sense since was only two tests. Overall, this section is content heavy and the reader needs a clearer path to follow the important discussion.

Related to Section 2.3, it is really hard to understand what figures 4 and 5 are trying to show. How does figure 4 show us that there is no contamination? What does the $CO_2$ and $N_2O$ tell us in 5a? This figure and/or the relevant text to it needs to be reconsidered so the reader doesn't need to spend the majority of their time trying to understand it.

To enhance the clarity of Section 2.3 and to better convey the results of each test and their implications, we have re-organized this section. We have also highlighted the significance of each figure in the main text to facilitate easier comprehension of the content. First, we flushed the AirCore zero-grade air, then stored overnight (~14 hours), and subsequently analyzed using the StratoCore-GC-ECD system, during which a standard gas of typical tropospheric composition was used as the push gas following the stored sample. Figure 4A shows the results: the first 164 mL of injection, corresponding to the gas inside the AirCore, we can see that the dry mole fractions of all target molecules were below the StratoCore-GC-ECD detection limit for these species in the entire, suggesting during the 14 hours storage did not have any contamination. Figure 4B was similar, the only difference is that we flushed the AirCore with standard gas, to investigate if the AirCores can act as sink of some molecules. Our results again showing that the inside of AirCore will not absorb our target molecules.

Section 3 and Figure 6:  Is there a reason why the parachute is red? What is the 'lightweight material'? Is the fact the balloon is off-white important?

Thank you for the comments. We revised Figure 6, deleted irrelevant information on the figure to make it more concise and easier to follow.

Line 268: "data not shown in figure" – Could the comparison between model and observed be done in a supplement if no space in the paper? It is referenced in the conclusion that there is good agreement between model and observations but there is nothing that directly shows that for the reader. The RMSE is given but more information would be nice.

Thank you for the comment. We agree that the comparison is useful for readers to better understand the performance of the model. Since this model was discussed in depth in Tans (2022), we originally did not include this part into our manuscript. Here we are attaching the model-measurement comparison in this document.

[Figure]

Lines 275-289: Vertical references are given as hPa but the plots are shown as km which is challenging to compare the figure to the text. Perhaps add a second y axis to Figure 9 with hPa?

Thank you for the comment. We agree that using hPa in the text but km in the figures can be confusing. Therefore, in the main text we replaced "hPa" to "km" to avoid such ambiguity.

Technical Corrections:
Line 20: We then launched
We have corrected this grammar error.

Line 44: chlorofluorocarbonsmolecules
We corrected this typo.

Line 114: add degree symbol to 38C
We corrected this typo.

Line 255: "after the descent" – during the descent? After the initial descent? The phrasing as written implies once it is on the ground
We changed this phrase to "after the balloon cutaway" to clarify.

Line 299: in Kansas
We corrected this grammar error.

Line 301: Are you saying the StratoCore data can get higher than the ER-2 or that it covers more of the stratosphere than the ER-2?
Here we are trying to express that the StratoCore sample can reach 25-28 km ASL, above ER-2 altitude, meaning it may cover more of the stratosphere in a cost-effective method. The two sentences here are repetitive so we deleted one of the sentences.

Figure 9: Is it possible to outline the symbols to make them easier to see? Or make them larger like in Figure 10.
We revised Figure 9, increased the size of the markers so they are easier to see. In addition, we added 3 panels in Figure 9 showing the vertical profiles of $CO_2$, CO and $CH_4$ measured on the four flights to provide more information.

---

## Author Comment (AC2)

This manuscript describes a neat and low-cost new development in characterising the trace gas composition of the stratosphere at an expanded range of species, while at the same time maintaining a high vertical resolution. The methods are well presented and have been thoroughly tested. My main concern is not with this new methodology itself, but with the detection method. It has been known for decades that ECDs are prone to interferences as they are sensitive to the dozens of chlorinated, brominated, and iodinated trace species in the atmosphere. This becomes very much a problem when compressing species with such a large range of boiling points into a 2-minute chromatogram - not so much for the low-boiling end (here: SF6, N2O) and CFC-12, but very much for H-1211, CFC-11, and CFC-113. This is completely ignored here and no evidence is provided that interferences have been identified (e.g., CFC-114 & -114a are know to elute at similar times as H-1211) or quantified. The Hintsa et al. (2021) paper that is cited does not disuss this problem either - but then the method is essentially the same (GC-ECD-based) and the paper also does not show plots of the problematic species. This interference problem might not be so pronounced in the troposphere, where none of these gases are photolysed. However, this changes very much in the stratosphere, especially as their interferents are photolysed at different rates. I therefore urge the authors to at least discuss these serious limitations, and ideally assess their influence on the results as well. Further specific and mostly minor comments can be found below.

We thank the reviewer for their comments to improve this manuscript and acknowledge the potential for co-elution interference in GC analysis. Co-elution interference in GC analysis is a global concern, as new species grow and shrink in the atmosphere, we all need to be cautious about this. One strength of our AirCore program is that it has strong ties to our surface network program where we run both ECD and MS chromatography. Regular intercomparisons between each of these different techniques could reveal these interferences. These intercomparisons are possible at the sub 1% level because of our common in-house standards program.

We would also like to address the the potential CFC-114 interference. Our flask program runs a near identical GC-ECD channels with a much slower GC-ECD analysis, despite differences in flow rates between instruments, the peak location in our StratoCore GC should be similar. The attached figure below shows the resolved CFC-114 peak between CFC-12 and H-1211. We see this peak in our StratoCore GC chromatography and can estimate that the contribution under the H1211 peak contributes less than 1 %.  In addition, CFC-114 has a much longer lifetime than H-1211 but our analysis shows that the mole fractions of H-1211 decreased to 0 at about 20 km AMSL, further suggesting that CFC-114 has not affected the H-1211 measurement on the StratoCore-GC-ECD. In addition, our "slow" flask GC-ECD measurement compares well with GC-MS analysis, which is much less impacted by this interference, allowing us to better investigate such discrepancies if they occur in the future.

[Figure]

l9-10 2 x "atmosphere in the first sentence.
We removed the 2nd "atmosphere".

l31 Throughout the manuscript: Please consider a consistent ordering when citing multiple references, e.g. alphabetically or by publication year.
We have revised the citation ordering in the manuscript to be order by publication year.

l36-50 This discussion is not well balanced as it ignores recent work on the lifetime of SF6 (e.g. Ray et al., JGR, 2017, Leedham Elvidge et al., ACP, 2018, and Loeffel et al., ACP, 2022), which provides strong evidence that this compound might not be completely inert and therefore not a direct age tracer.
Thank you for the comment. We revised this part of the introduction to point this out.  The mesospheric SF6 loss measured and presented in Ray et al. suggested a small 3 month correction for AoA in the mid-latitudes at 32 km in 2000 and was just starting to approach to the detection limit of AoA measurements at that time.  This loss, however, is proportional to the value of $SF_6$. Taking in to account the accompanying increase in growth rate the correction now is approximate 2.7 times larger and must be accounted for when using SF6 for AoA.  The measured $SF_6$ mole fraction in the midlatitudes now contains measurable information not only about AoA but also the mass exchange between the stratosphere and the mesosphere, which was only obtainable in Vortex profiles before. We see this as a gain not a loss. Especially since both CO2 AoA and SF6 AoA will be measured.

l51 Tans, 2009 is not in the reference list.
We added Tans 2009 to the reference list.

l81-82 It is not clear what the "ten stratospheric measurements" mean. Are these ten samples from one flight or ten flights? Also, why not state the actual published number instead of saying "around"?

We clarified this sentence: ten stratospheric measurements from a 2 L volume AirCore can be measured in each flight.

l82 No 2 L AirCores were used for the Laube et al., 2020 paper.

We have revised this part to avoid future confusion.

l90 Please clarify that the top 25 % and the stratospheric portion are not the same thing as tropopause pressures can vary substantially with season and location.

We changed the statement to "stratospheric portion of AirCore samples (approximately the first 20%-30% of the sampler tubing)" to acknowledge the fact that the location of tropopause can vary.

l94-101 Please shorten to avoid the repetition of the (very valid) point on the advantages of "high resolution".

We revised this part of the manuscript to avoid repetition of "high resolution".

l127 Looking at Figure 9, SF6 (as expected) does not vary by 50-100%.

We clarified this statement: 50%-100% overall variations for CFCs and $N_2O$, and 20% for $SF_6$.

l128-130 When comparing Figure 3 and Figure 9, it is apparent that the observed stratospheric mole fraction range extends well below the range for which the GC-ECD response behaviour was characterised. The effects on the analytical uncertainties of such low mole fractions, and the resulting limitations on constraining circulation changes should at least be discussed in a qualitative manner.

Some observed CFC-11 mixing ratios are indeed outside the range of the calibration, however, the response function is well-described by a $2^{nd}$ order polynomial (Fig. 3). We feel that extrapolation below the lowest calibration standard does not introduce significant additional uncertainty. Further, the GC-ECD response to zero air identifies the curvature across the integrated peak region and is used to validate the zero intercept point on the calibration curves.

l138-140 At what pressure is that cylinder gas being pushed through? Would this induce extra mixing?

We move the sample out of AirCore using a small push flow of 4-5 mL/min that is strictly controlled by a mass flow controller to ensure stability thereby minimizing pressure fluctuations. It generates a few psi of pressure gradient across the entire AirCore. This slow flow minimizes the mixing between push gas and the neighboring sample gas, and between neighboring AirCore samples. This push flow mixing has been verified by laboratory tests, as shown in Figure 4A & 4B.  We would also like to point out that we analyze air from the top of the profile first and no push flow induced mixing exist here. As we continue pushing the air off the AirCore the flow induced mixing increases to its maximum value at our last data point, by then the altitude to data point resolution is much better and can handle the small amount of

flow induced mixing between data points that has accumulated.  We have revised the paragraph accordingly to clarify this.

l142-147 Please quantify "carefully controlled". What flow rates are being used, what are the related uncertainties, and how does this translate into sample volume uncertainties?

We clarified our statement to better describe the flow controlling system and flow measurement system. The mass flow controller will control the flow to a pre-set value (usually 4-5 mL per minute), to ensure a stable pressure in the system but the actual flow rate (which is crucial for calculating sample volume) is measured by another mass flow meter. The flow measurement by mass flow meter is accurate within 0.6%, as discussed in section 2.3.

l149-159 This is a very nice experiment. However, two questions that are not addressed (and which might provide limitations to the conclusions drawn) are 1) According to Karion et al., 2010 the AirCore also usually contains a magnesium prechlorate drier. Was that also tested? And 2) Were these test carried out at the temperatures that AirCores cool down to during actual balloon flights?

To answer the questions: 1) we did not include a drier on the AirCore for GC-ECD analysis, since it may potentially contaminate the CFCs. However, the degree of contamination may need further investigation. 2) The tests are carried out at room temperature (~293 K). During the flights, the temperature of AirCore tubing usually range from 263K to 300K due to the insulation of the AirCores, although the ambient temperature can be as low as ~210K. Since the AirCore temperature were relatively stable during flights, we suggest the tests in this manuscript can represent the AirCore condition in actual flights.

L181 If the chromatogram, as indicated in Figure 2, is 2 mins long, the size of each sample is 4-5 ml (l92), and about 250 ml of air are analyzed (l136), this gives a minimum time of 100 minutes for the analysis of the upper part of an AirCore, not including any flushing, backflushing or calibration standard measurement times. This seems to be inconsistent with this experiment only taking 1 hour, unless it was carried out at higher flow rates (which would make it less representative of an actual flight).

In the experiment we used a smaller AirCore (164 mL as shown in Figure 4A & 4B), so the 1 hour used here is the time it takes to analyze this small AirCore. We clarified this in the manuscript. For actual AirCore flights, the analysis time is approximately 2 hours. Following Eq. 1, we calculated the mean diffusion distance to be ~52.9 cm, corresponding to ~3 mL of air. Again, we analyzed air from the top of the profile first where diffusion is at its minimum. As we continue analyzing the AirCore thermal mixing increases to its maximum value at our last data point, but by then the altitude to data point resolution is much better and can better handle the effect of thermal mixing on the data set. The quality of the data set is still dominated by the thermal mixing that occurs between filling the AirCore at altitude and recovery time needed to start the analysis.

l264 It looks like like "between" is missing after "imbalance".

We have fixed this error.

l264 It is not clear how this pressure imbalance was measured.
We added a sentence here to describe how we measured the pressure imbalance: we mounted a pressure transducer on the closed end of the AirCores, which measures the pressure differences between the closed end of the AirCore and ambient air. Figures were added to reviewer comments above showing the measured vs. modeled differential pressure of the AirCores during flights 1-4.

l283 Please indicate the approximate altitude of the "650 K isentrope" or, alternatively, add an explanation of why this coordinate was used here.
We added an approximate altitude of the isentrope.

l285 These values only "agree well" qualitatively. Also, please provide a reference for the expected photolysis rate order.
We revised the statement "agree well" to "qualitatively agree" and added a reference.

l288 It is not made clear to the reader, exactly how the authors derived that "variability on scales of days to weeks" was captured here.
We speculate the observed excursions in our data are temporary structures in the lower stratosphere. To avoid confusion, we deleted the "scales of days to weeks".

l289-290 It seems like a missed opportunity not to show the CO2 and CH4 results as well. Why would you fly two AirCores alongside each other and then only display the profiles from the new, but not those from the established method? This is especially apparent here, where the latter results are discussed, but the reader left in the dark on how well these "similar structures actually" agree with each other.
Although this paper mainly introduces the StratoCore-GC-ECD analytical method, we agree that showing the results of our "traditional" continuous flow analysis method using a cavity ring-down spectrometer for measurement of CO2, CH4 and CO is helpful. We added 3 panels on Figure 9 showing the CO2, CH4 and CO results. The temporal stratospheric variability, such as the variable mole fractions of trace gases at 10-17 km of Flight 3, is not only observed from the CFCs, but also shown in CO2, CH4 and CO.

l313-317 It is not clear to me, how the authors arrived at this conclusion. Looking at Figure 9, the lowest SF6 mole fraction also appears to have been measured at Flight 2.
By "lowest SF6 mole fraction" we meant the samples with highest altitude of Flight 3 (thus the lowest SF6 mole fractions). The SF6-N2O relationship of these data points seem to deviate from other observations (including DCOTTS data and other AirCore data), therefore we speculate that this deviation is originated from some short term stratospheric variabilities. We have revised this part to avoid future confusions.

---

## Author Comment (AC3)

Overall Summary:

This paper is within the scope of AMT and should be published. There are some confusing sections, discussed further below, but overall the paper does a nice job of explaining a new technique that will improve our understanding of stratospheric circulation and add important measurements for long-term climate relevant gases. They prove that this technique works well and appears easy to add on to already existing infrastructure, and is cost-effective. There is nice agreement with aircraft measurements and any discrepancies are reasonably explained. I look forward to seeing more StratoCore-GC-ECD data in the future. Given that only one section of this paper needs restructuring, it should be accepted with minor revisions.

General comments:

The main section of the paper that needs improvement is section 2.3 and relevant figures. Section 2.3 was the most confusing of the entire paper and also critical to the verification of this new system. It was hard to follow the steps of the different experiments. In part this was a challenge because of the different language used (test, experiment, etc) along with the two tests (but maybe more?) to study three scientific objectives. This section is important and a restructure would be helpful for the reader to follow the logic. For example, line 160 mentions 'both experiments' and then line 192 mentions 'another set of tests'. Was that related to the first two tests mentioned at the start of the section? If so, set doesn't make sense since was only two tests. Overall, this section is content heavy and the reader needs a clearer path to follow the important discussion.

Related to Section 2.3, it is really hard to understand what figures 4 and 5 are trying to show. How does figure 4 show us that there is no contamination? What does the $CO_2$ and $N_2O$ tell us in 5a? This figure and/or the relevant text to it needs to be reconsidered so the reader doesn't need to spend the majority of their time trying to understand it.

Thank you for the comment. To enhance the clarity of Section 2.3 and to better convey the results of each test and their implications, we have re-organized this section. We have also highlighted the significance of each figure in the main text to facilitate easier comprehension of the content. Figure 4A and 4B shows the results from two experiments. In the first experiment, we flushed the AirCore zero-grade air, then stored it overnight (~14 hours), and subsequently analyze it using the StratoCore-GC-ECD system, during which a standard gas of typical tropospheric composition was used as the push gas following the stored sample. The results in Figure 4A shows that, the first 164 mL of injection, corresponding to the gas inside the AirCore, the dry mole fractions of all target molecules were below the StratoCore-GC-ECD detection limit for these species. This suggests that during the 14 hours storage, the AirCore tubing did not induce any contamination. Figure 4B was similar, the only difference is that in this experiment, we flushed the AirCore with standard gas, to investigate if the AirCores can act as sink of some molecules. Our results again showing that the inside of AirCore will not absorb any of our target molecules.

Section 3 and Figure 6: Is there a reason why the parachute is red? What is the 'lightweight material'? Is the fact the balloon is off-white important?

Thank you for the comments. We revised Figure 6, deleted irrelevant information on the figure to make it more concise and easier to follow.

Line 268: "data not shown in figure" – Could the comparison between model and observed be done in a supplement if no space in the paper? It is referenced in the conclusion that there is good agreement between model and observations but there is nothing that directly shows that for the reader. The RMSE is given but more information would be nice.
Thank you for the comment. We agree that the comparison is useful for readers to better understand the performance of the model. Since this model was discussed in depth in Tans (2022), we originally did not include this part into our manuscript. Here we are attaching the model-measurement comparison in this document.

[Figure]

Lines 275-289: Vertical references are given as hPa but the plots are shown as km which is challenging to compare the figure to the text. Perhaps add a second y axis to Figure 9 with hPa?
Thank you for the comment. We agree that using hPa in the text but km in the figures can be confusing. Therefore, in the main text we replaced "hPa" to "km" to avoid such ambiguity.

Technical Corrections:
Line 20: We then launched
We have corrected this grammar error.

Line 44: chlorofluorocarbons**molecules**

We corrected this typo.

Line 114: add degree symbol to 38C

We corrected this typo.

Line 255: "after the descent" – during the descent? After the initial descent? The phrasing as written implies once it is on the ground

We changed this phrase to "after the balloon cutaway" to clarify.

Line 299: in Kansas

We corrected this grammar error.

Line 301: Are you saying the StratoCore data can get higher than the ER-2 or that it covers more of the stratosphere than the ER-2?

Here we are trying to express that the StratoCore sample can reach 25-28 km ASL, above ER-2 altitude, meaning it may cover more of the stratosphere in a cost-effective method. The two sentences here are repetitive so we deleted one of the sentences.

Figure 9: Is it possible to outline the symbols to make them easier to see? Or make them larger like in Figure 10.

We revised Figure 9, increased the size of the markers so they are easier to see. In addition, we added 3 panels in Figure 9 showing the vertical profiles of CO2, CO and CH4 measured on the four flights to provide more information.